# Structure, Spectra, Morphology, and Magnetic Properties of Nb⁵⁺ Ion-Substituted Sr Hexaferrites

**Wenhao Zhang [1,2,\*], Pengwei Li [1], Yonglun Wang [1], Jing Guo [1], Jie Li [1,\*], Shuo Shan [1], Saisai Ma [1] and Xing Suo [1]**

1 Key Laboratory of Integrated Exploitation of Bayan Obo Multi-Metal Resource, School of Materials and Metallurgy, Inner Mongolia University of Science and Technology, Baotou 014010, China; li1027623129@163.com (P.L.); wy9960209sf@163.com (Y.W.); inner960613@163.com (J.G.); zyn20210321@163.com (S.S.); xncd0221@163.com (S.M.); xshi20201020@163.com (X.S.)

2 School of Mining and Coal, Inner Mongolia University of Science and Technology, Baotou 014010, China

\* Correspondence: zhang1949china@163.com (W.Z.); yjslijie@126.com (J.L.)

**Abstract:** $SrFe_{12-x}Nb_xO_{19}$ (x = 0.00–0.15) was here synthesized by a conventional solid-state reaction method. Thermogravimetry and differential scanning calorimetry curves revealed the sample reactions at four temperature ranges, and the optimal reaction stability was obtained at 1240 °C. A single-phase polycrystalline form of $SrFe_{12}O_{19}$ was obtained until the substitution reached 0.09, and the average crystallite size was found to be in the range of 44.21–60.02 nm. According to Fourier-transform infrared spectra, the formation of Fe–O bonds occurred at 69 and 450 cm$^{-1}$ in the M-type ferrite, while Raman spectra revealed that all the peaks in the sample corresponded to Raman vibration modes and M-type structures. Through the shift of the peaks, it is speculated that $Nb^{5+}$ enters into the lattice. The hysteresis loops of the samples were measured by vibrating-sample magnetometry, and the calculated results demonstrated that the coercivity increased with increases in the doping amount (686.3 Oe). At the same time, the saturation magnetization remained at a large value (>72.49 emu/g), which has rarely been reported.

**Keywords:** M-type hexaferrite; spectrum; morphology; magnetic measurements

## 1. Introduction

M-type strontium ferrite magnetic materials are widely used in permanent magnets, magnetic recording, microwave devices, loudspeakers, communication, automobile, motor, and other fields because of their high coercivity, good saturation magnetization, low dielectric loss, good oxidation resistance, good stability, and relatively simple manufacturing process [1,2]. In M-type hexaferrites, $Fe^{3+}$ ions are distributed among five crystallographic sites: one trigonal-bi-pyramidal site (2b↑), one tetrahedral site (4f1↓), and three octahedral sites (12k↑, 2a↑, and 4f2↓) [3]. The properties of ferrite also depend on the microstructure. Ion substitution is a common method to adjust the properties based on alteration of the crystal structure, such as by La [4], Al [5], Mn [6], Cr [7], or Nd [8], or multiple ion substitution, such as of La–Zn [9], Co–Zr [10], Pr–Zn [11], Bi–La–Y [12], or Bi–Co–Ti [13]. Previous results have demonstrated that the properties of the composites can be improved and reduced in different degrees to meet the needs in various fields. The most common preparation methods include sol–gel processes [13], conventional solid-state routes [14,15], co-precipitation methods [16], hydrothermal methods [17], and ball milling methods [18].

However, the literature review shows that there are a few articles on the preparation of M-type strontium ferrites by Nb doping. Fang et al. [19] reported a paper entitled "Effects of Zn–Nb Substitution on Magnetic Properties of Strontium Hexaferrite Nanoparticles", Yang et al. [20] reported a paper titled "The Influence of Nd-NbZn Co-substitution on Structural, Spectral and Magnetic Properties of M-type Calcium-Strontium Hexaferrites $(Ca_{0.4}Sr_{0.6-x}Nd_xFe_{12.0-x}(Nb_{0.5}Zn_{0.5})_xO_{19})$", and Güner et al. [21] reported a paper entitled "Microstructure, Magnetic and Optical Properties of $Nb^{3+}$ and $Y^{3+}$ Ions Co-substituted Sr

hexaferrites". However, no one has reported on the single-doping of Nb elements. In this paper, we report the successful preparation of $SrFe_{12-x}Nb_xO_{19}$ by Nb doping on the Fe site for the first time.

## 2. Experimental Details

### 2.1. Materials

Iron oxide ($Fe_2O_3$; 99.7%), strontium carbonate ($SrCO_3$; 99%), and niobium oxide ($Nb_2O_5$; 98%) were used as starting materials without any treatment.

### 2.2. Synthesis of M-Type Hexaferrites

According to the $SrFe_{12-x}Nb_xO_{19}$ stoichiometric ratio of ingredients, a ball-to-powder ratio of approximately 10:1 was placed into a nylon jar containing a certain proportion of ultrapure water. It was then wet milled for 5 h, after which the planet was removed from the nylon grinding jar and left at 100 °C to dry in the bellows. Then, it was filtered for zirconia ball and powder separation, yielding a round powder under 15 MPa of pressure. Next, it was sintered in a tube furnace at a heating rate of 5 °C/min from room temperature to 1240 °C. It was held at this temperature for 2 h and then cooled via furnace cooling.

### 2.3. Characterizations

The thermal behavior of the sample was examined by thermogravimetry and differential scanning calorimetry (TG–DSC; STA 449C, Netzsch, Hanau, Germany) at a temperature range of room temperature up to 1300 °C with a heating rate of 10 °C/min. The X-ray diffraction (XRD) patterns of the calcined magnetic powders were obtained by an X-ray diffractometer (Rigaku smart lab, Tokyo, Japan) in continuous mode using a CuK$\alpha$ radiation source ($\lambda$ = 1.5406 Å), and the 2$\theta$ angles were scanned over a range from 20° and 70° at equal steps of 0.02°. Infrared spectral analysis was performed using Fourier-transform infrared spectroscopy (FTIR; VERTEX 70, Bruker, Berlin, Germany) within a wave number range of 2000–400 cm$^{-1}$. The Raman spectra were collected with a Renishaw inVia-Qontor Raman spectrometer (Renishaw, Stroud, UK), using a wave number range from 100 to 800 cm$^{-1}$ within a single spectral window. Field emission scanning electron microscopy (FE-SEM; Supra 55, ZEISS, Oberkochen, Germany) was used to analyze the particle size and surface morphology of the samples. The room temperature magnetic properties of all samples were investigated using vibrating-sample magnetometry (VSM; Versa Lab, Quantum Design, Beijing, China), with a maximum field of 30 kOe.

## 3. Results and Discussion

### 3.1. Thermal Analysis

It can be seen from Figure 1 that the TG–DSC curve as a whole could be divided into four reaction stages. The first stage is 23–611 °C, where the TG curve is close to being flat, and there is no obvious drop. The second stage is 611–837 °C, where the weight loss was as high as 3.43%, because the $SrCO_3$ in the material was decomposed by heat. It can be seen from the DSC curve that there were two obvious exothermic peaks from 837 °C to 907 °C, which may have been the generation of $SrFeO_{3-x}$ phase. As well as the formation of M-type strontium ferrite, this special magnetic phase transition represents the transition from ferromagnetism (FM) to paramagnetism (PM), and the temperature is defined as the Curie temperature (Tc) [22]. The third stage was 849–1240 °C. Here, $SrCO_3$ and $Fe_2O_3$ were heated to form an intermediate product, $SrFeO_{3-x}$, and the transition process to $SrFe_{12}O_{19}$ took place as the temperature rose. Finally, the final product was formed at 1240 °C, at which the fourth stage began. The curve during this stage tended to be flat without obvious fluctuations, indicating that the reaction was complete.

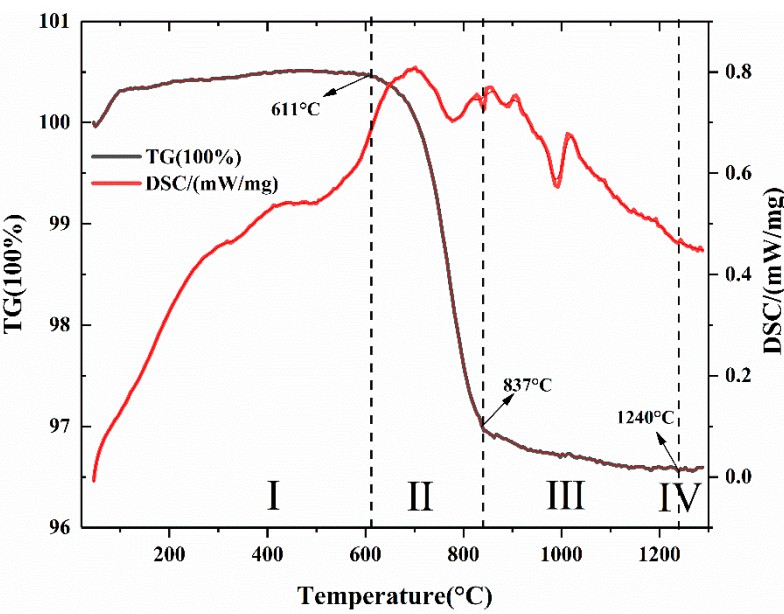

**Figure 1.** TG/DSC scanning curve for $SrFe_{12}O_{19}$.

### 3.2. X-ray Diffraction Analysis

The powder XRD patterns of the $SrNb_xFe_{12-x}O_{19}$ (x = 0.00–0.15) hexaferrites are given in Figure 2. Comparing all the XRD patterns with the standard M-type strontium ferrite PDF card (PDF-33-1340), it was found that when x < 0.09, single-phase $SrFe_{12}O_{19}$ could be obtained. When x $\geq$ 0.09, an impurity peak appeared at $2\theta$ = 33.16° ($\alpha$-$Fe_2O_3$), and the intensity of the peak gradually increased with increases in the doping amount. It can be seen that when the Nb was excessively doped, the iron phase precipitated, and it was impossible to synthesize pure M-type strontium ferrite. In summary, when x < 0.09, a relatively pure M-type hexagonal ferrite phase could be obtained.

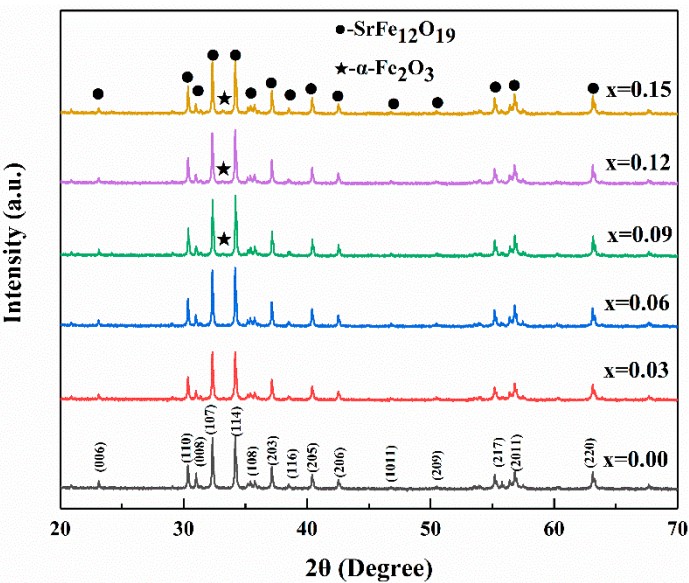

**Figure 2.** Powder XRD patterns of $SrFe_{12-x}Nb_xO_{19}$ (0.00 $\leq$ x $\leq$ 0.05) HFs.

Figure 3 gives the Rietveld refinement of all diffraction peaks for the sample at room temperature. As a rule, the whole samples were refined based on the corresponding structure with the space group P63/mmc [23], and all the parameters are listed in Table 1. From the table, it can be seen that the lattice constant *a* did not change significantly with the increase of Nb, while the lattice constant *c* tended to increase with the increase of

the substitution amount. However, the range of this change was small, because after $Nb^{5+}$(0.64 Å) replaced $Fe^{3+}$(0.64 Å), the radius did not undergo much alteration [1]. The average crystallite size (*D*) was calculated by the Debye–Scherrer formula:

$$D = \frac{k\lambda}{\cos\theta} \tag{1}$$

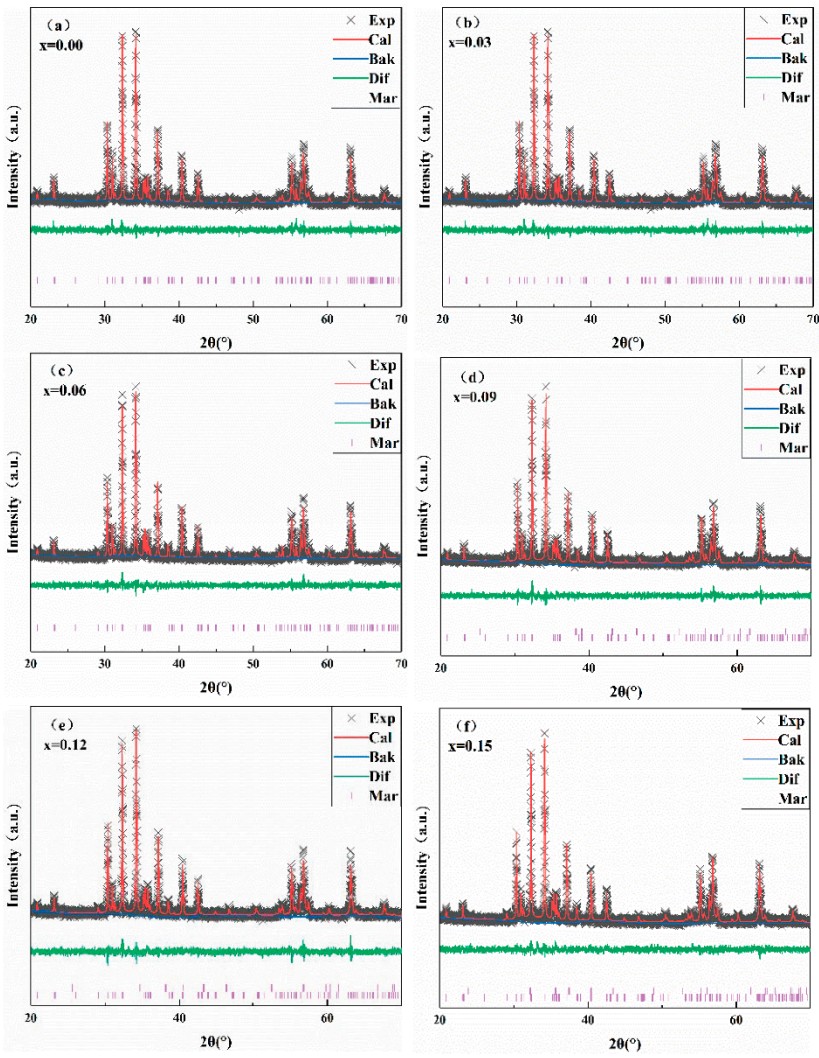

**Figure 3.** Rietveld refined power XRD patterns for all Nb substituted hexaferrites. (**a**) x = 0.00; (**b**) x = 0.03; (**c**) x = 0.06; (**d**) x = 0.09; (**e**) x = 0.12; (**f**) x = 0.15.

**Table 1.** Lattice parameters (*a*, *b*, and *c*; in Å), *c*/*a* ratio, unit cell volume ($v_{cell}$; in Å), and $\chi^2$ for all $SrFe_{12-x}Nb_xO_{19}$ (x = 0.00–0.15) hexaferrites.

| x | *a* (nm) | *c* (nm) | *c*/*a* | $v_{cell}$ | *D* (nm) | $\chi^2$ |
|---|---|---|---|---|---|---|
| x = 0.00 | 5.891 | 23.079 | 3.917 | 693.63 | 46.20 | 1.39 |
| x = 0.03 | 5.886 | 23.058 | 3.917 | 693.54 | 45.18 | 1.16 |
| x = 0.06 | 5.887 | 23.060 | 3.917 | 692.02 | S | 1.14 |
| x = 0.09 | 5.884 | 23.067 | 3.920 | 693.27 | 47.66 | 1.20 |
| x = 0.12 | 5.884 | 23.069 | 3.921 | 690.72 | 49.24 | 1.23 |
| x = 0.15 | 5.886 | 23.082 | 3.922 | 691.01 | 44.21 | 1.14 |

The calculation results of the average crystal grain size (*D*) are listed in Table 1. The average crystallite size was less than 50 nm, which is the requirement to obtain a suitable

signal-to-noise ratio for high-density recording media [24]. When x = 0.06, the grain size $D > 50$ nm, which could be the reason for the experimental error. The cell volume also tended to decrease with the increase of the Nb doping amount. This is because the difference in radius caused by ion substitution made the *a*- and *c*-axes undergo distinct alterations. $\chi^2$ is the refined structure parameter derived by the Rietveld method (<10 is reasonable). The small value indicates that the optimized extraction value is very close to the real value of the sample, and the synthesized sample is very good.

It can be seen from Table 1 that when x < 0.12, the value of *c/a* was between 3.91 and 3.92, which also strongly indicates that the obtained sample was M-type ferrite [25] (x = 0.15, *c/a* = 3.921). Thus, it was inferred that with the excessive increase of Nb element doping, the M-type ferrite structure began to undergo damage.

### 3.3. Fourier-Transform Infrared Spectroscopy Study

Figure 4 shows the FTIR spectra of different compositions of $SrFe_{12-x}Nb_xO_{19}$ (x = 0.00–0.15) hexaferrites recorded at room temperature within a wavenumber range of 400–2000 $cm^{-1}$. The two distinct absorption peaks at 450 and 619 $cm^{-1}$ represent the vibration of $Fe^{3+}-O^{2+}$ at a particular site, corresponding to the formation of tetrahedrons and octahedrons, respectively, and indicating the formation of the lattice and structure of M-type SrLaM hexagonal ferrites [26]. No other characteristic bands were present, confirming that all the organic compounds were entirely eliminated [27].

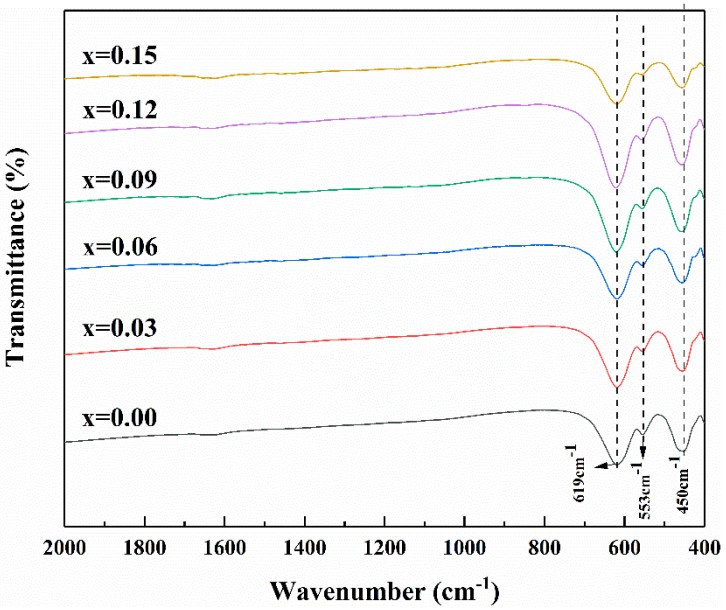

**Figure 4.** FTIR spectra of $SrFe_{12-x}Nb_xO_{19}$ (x = 0.00–0.15) hexaferrites.

### 3.4. Raman Analysis

**Raman spectroscopy is an important characterization method for analyzing the structure of M-type hexagonal ferrite. It is of great significance for studying the lattice distortion and crystallite size of materials. It can be seen from Figure 5 that all the characteristic peaks correspond to M-type hexagonal ferrite, which is consistent with the XRD test results.** The results indicated that there were 42 Raman-active modes ($11A_{1g} + 14E_{1g} + 17E_{2g}$), 30 IR-active modes ($13A_{2u} + 17E_{1u}$), and 54 silent modes ($3A_{1u} + 4A_{2g} + 13B_{1g} + 4B_{1u} + 3B_{2g} + 12B_{2u} + 15E_{2u}$) [28]. It can be seen from the figure that as the amount of doping increased, all peaks were shifted in the lower-wave-number direction. This is because the level of $Nb^{5+}$ substitution was increased, resulting in the displacement of oxygen atoms. Accordingly, the vibration energy was changed, which caused the peaks to shift, also confirming that Nb was incorporated into the strontium ferrite lattice. The intensity of the Raman peak also increased with the increase of the doping amount, which may

have been due to the lattice strain occurring in the lattice position. This can be confirmed in the XRD pattern, while the lattice constant *c* increased with the increase of the doping amount [29]. The bond at 409 cm$^{-1}$ was assigned to $A_{1g}$ vibration at the 12k sites, while that at 528 cm$^{-1}$ occurred owing to $E_{1g}$ vibration of the Fe–O bond. The band at 615 cm$^{-1}$ may have been the result of $A_{1g}$ vibration at the $4f_2$ site, while the bond at 685 cm$^{-1}$ was reflective of $A_{1g}$ vibration at the $4f_1$ sites. It has been previously reported that $Nb^{5+}$ can exist at the $4f_2$ and 12k positions. The peak displacement confirms that $Nb^{5+}$ entered the crystal lattice and provides theoretical support for the changes in magnetic properties outlined below [30].

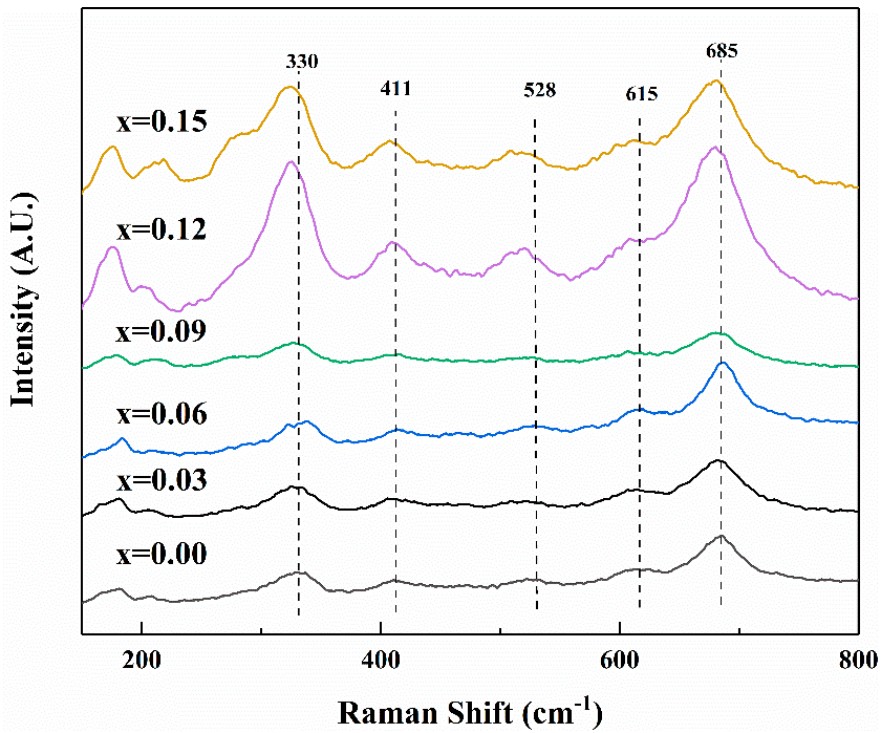

**Figure 5.** **Raman spectra of the** $SrFe_{12-x}Nb_xO_{19}$ (x = 0.00–0.15) **powders**.

*3.5. Morphological Analysis*

Figure 6 shows an FE-SEM image of $SrFe_{12-x}Nb_xO_{19}$ (x = 0.00–0.15) prepared by solid-phase sintering. Here, it can be seen that all samples had hexagonal grains approximately 1 μm in size. With the increase of Nb element, the microstructure became porous, and the grain size was refined (the grain size decreases from about 1 μm to 0.5μm), indicating that the average grain size and morphology were more sensitive to the doping of Nb element. This is because, after the entry of Nb ions, the change in internal stress caused distortion to the crystal lattice. To determine the chemical composition of the substance, $SrFe_{12-x}Nb_xO_{19}$ (x = 0.00–0.15) was characterized using energy-dispersive X-ray spectroscopy (EDS). Figure 7 shows the element distributions of strontium, iron, oxygen, and niobium when x = 0.09. Obviously, all the main elements of the sample were evenly distributed because there was no obvious difference in color spots in the EDS map, which appeared as a single color. The performance observed based on the map reveals that all of the SrM hexaferrite particles were doped with Nb [31]. The EDS test results show that with the increase of doping amount, the weight concentration of Nb in different samples continuously increases from 0 to 4.1%, indicating that the theoretical design was consistent with the actual material [32].

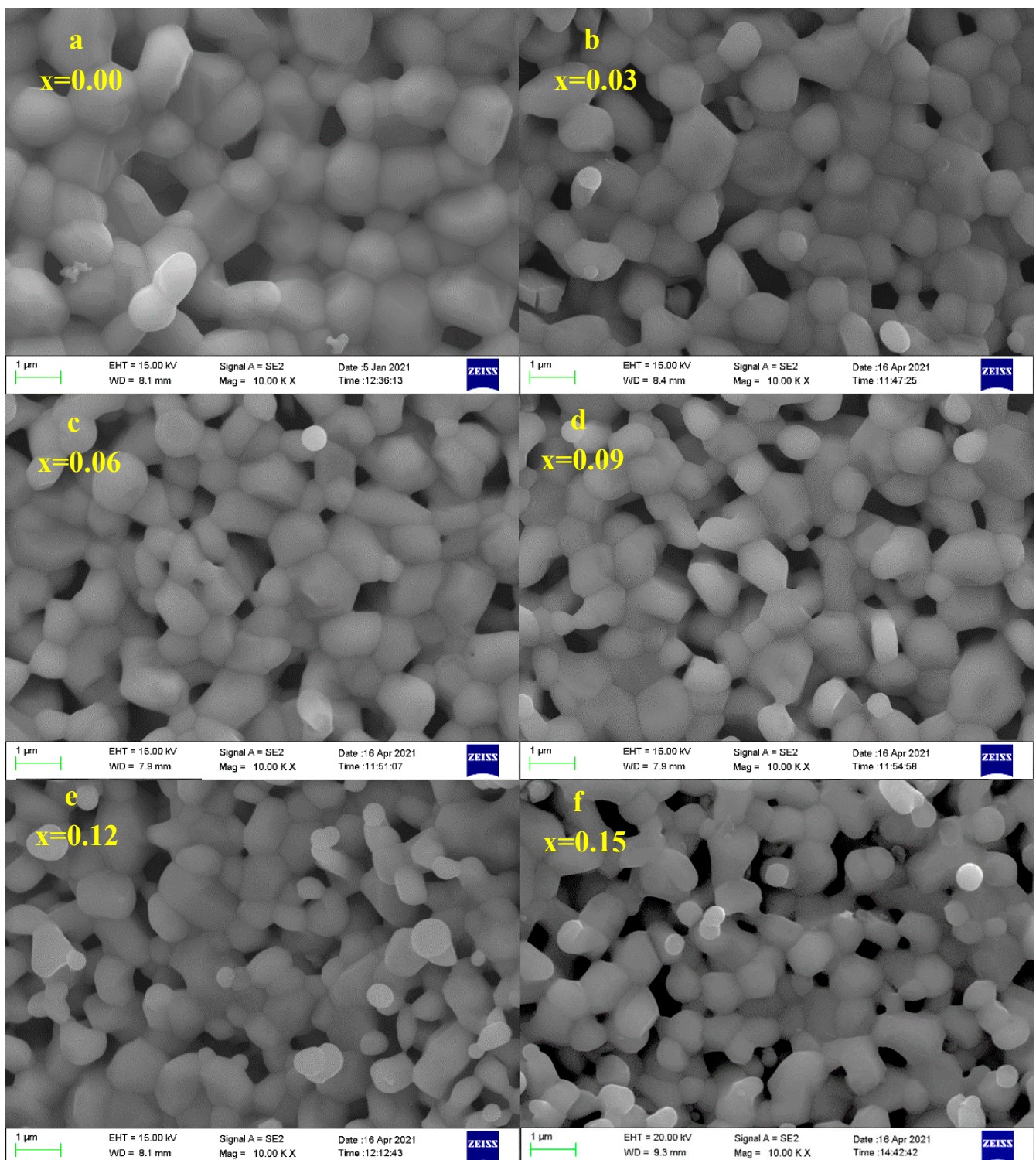

**Figure 6.** SEM micrographs of $SrFe_{12-x}Nb_xO_{19}$ (x = 0.00–0.15) hexaferrites. (**a**) x = 0.00; (**b**) x = 0.03; (**c**) x = 0.06; (**d**) x = 0.09; (**e**) x = 0.12; (**f**) x = 0.15.

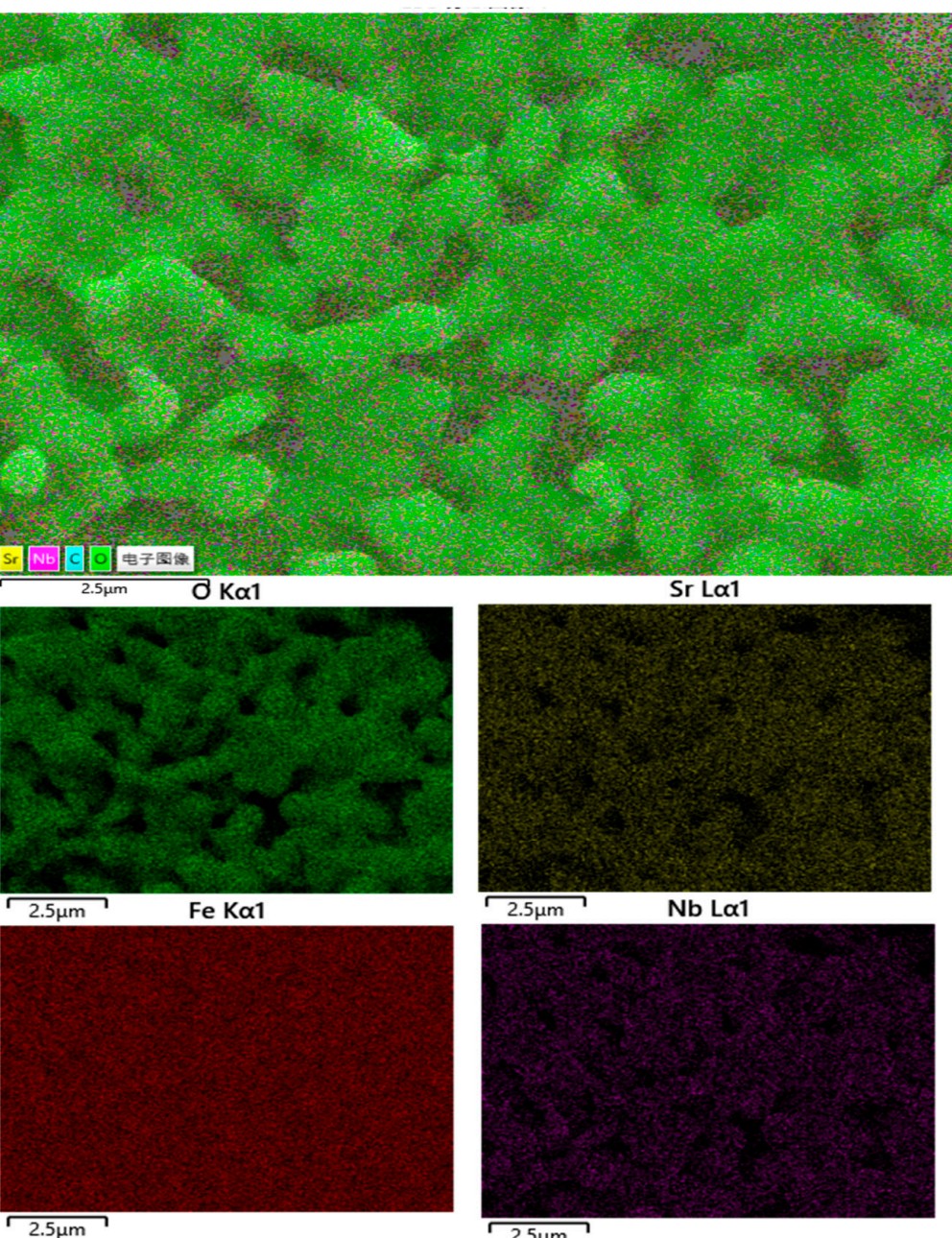

**Figure 7.** Elemental mapping results for SrFe$_{12-x}$Nb$_x$O$_{19}$ (x = 0.09) hexaferrites.

### 3.6. Magnetic Property Analysis

The magnetic characteristics of the synthesized samples were revealed using VSM. The M–H curve for SrNb$_x$Fe$_{12-x}$O$_{19}$ (x = 0.00–0.15) at an applied field value of $\pm$30 kOe is shown in Figure 8. The values of saturation magnetization ($M_s$), remanence ($M_r$), coercivity ($H_c$), squareness ratio ($M_r/M_s$), Bohr magneton number ($n_B$), anisotropy field ($H_a$), and anisotropy constant ($K_1$) are listed in Table 2. The magnetic nature of M-type hexaferrites depends on the electronic configurations and amount of the substituted cations as well as their site of preference, whether the tetrahedral site (4f$_1$), octahedral site (12k, 4f$_2$, and 2a), or trigonal bipyramidal site (2b) [19].

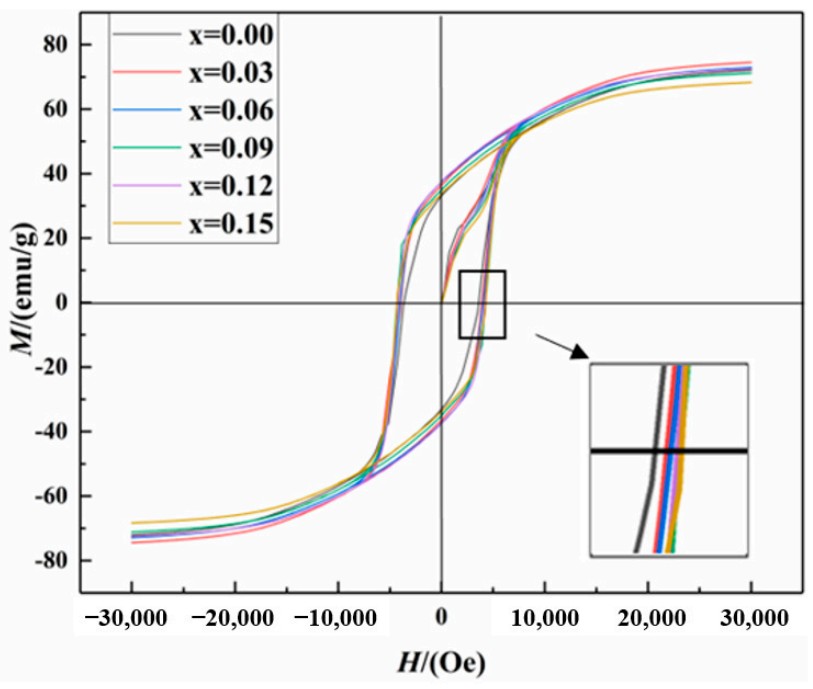

**Figure 8.** Magnetic hysteresis loops of $SrNb_xFe_{12-x}O_{19}$ (x = 0.00–0.15) hexaferrites.

**Table 2.** Magnetic parameters for $SrNb_xFe_{12-x}O_{19}$ (x = 0.00–0.15) hexaferrites.

| x | $M_S$/(emu/g) | $M_r$/(emu/g) | $M_r/M_s$ | $H_c$ (Oe) | $K_1$ ($\times 10^5$ emu/cm³) | $H_a$ (kOe) | $n_B$ |
|---|---|---|---|---|---|---|---|
| 0.00 | 74.68 | 33.52 | 0.449 | 3622.9 | 7.02 | 1.88 | 14.19 |
| 0.03 | 76.76 | 36.52 | 0.476 | 3926.1 | 6.77 | 1.76 | 14.61 |
| 0.06 | 75.24 | 37.59 | 0.499 | 4037.9 | 6.68 | 1.78 | 14.35 |
| 0.09 | 73.20 | 35.09 | 0.479 | 4277.3 | 6.12 | 1.67 | 13.96 |
| 0.12 | 74.49 | 37.79 | 0.507 | 4197.5 | 6.08 | 1.63 | 14.22 |
| 0.15 | 70.18 | 33.82 | 0.482 | 4309.2 | 5.59 | 1.59 | 13.41 |

As shown in Figure 9, the value of $M_s$ increased with the increase of x; when x < 0.06, which may be because $Nb^{5+}$ caused the crystal field of adjacent $Fe^{3+}$ to change after entering the crystal [33], increasing the ultrafine field of the 12k and 2b crystal sites. Eventually, the super-exchange effect was increased, thereby increasing the magnetocrystalline anisotropy field in the c-axis direction, which in turn increased the $M_s$. Liu [34] has proposed that the increasing $M_s$ may result from the valence change of $Fe^{3+}$ to $Fe^{2+}$ at the $4f_2$ site due to the reduction of the negative magnetic moment. When x > 0.06, it decreased with the increase of the doping amount. This is because the 2a, 2b, and 12k crystal positions of $Fe^{3+}$ rotated upwards, while the $4f_1$ and $4f_2$ positions rotated downwards. It has been reported that $Nb^{5+}$ can be at the $4f_2$ or 12k positions. When substituting $Fe^{3+}$ with $Nb^{5+}$, it is important to note that $Nb^{5+}$ is non-magnetic. The magnetic moment of $Fe^{3+}$ is 5 $\mu_B$, and the magnetic moment of $Nb^{5+}$ is 0 $\mu_B$, meaning magnetic dilution will occur. Therefore, the ion substitution leads to a decrease in the molar magnetic moment, causing $M_s$ and $M_r$ to decrease. In addition, as the doping amount increases (x > 0.06), the second term $\alpha$-$Fe_2O_3$ begins to appear, and the peak size gradually becomes larger as the doping amount increases. This is also one of the factors causing the decrease of $M_s$, while $M_r$ and $M_s$ exhibited the same change trend.

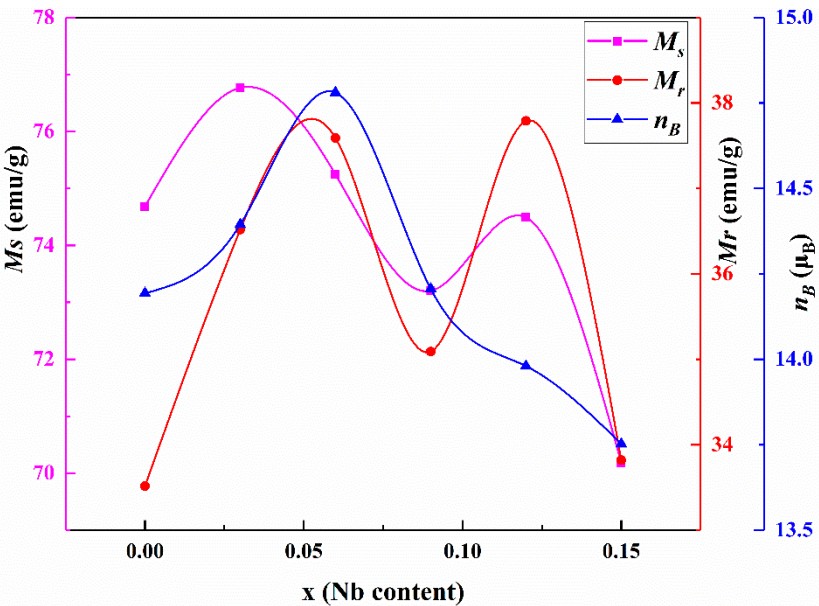

**Figure 9.** Variation of $M_s$, $M_r$, and $n_B$ with Nb substitution.

The value of $M_s$ can be calculated from hysteresis loops using the law of saturation [35]:

$$M = M_s\left(1 - \frac{b}{H^2}\right) \qquad (2)$$

Here, $M_s$ is the theoretical saturation magnetization, and $b$ is a constant related to the magnetocrystalline anisotropy.

As shown in Figure 10, the coercivity $H_c$ increased with the increase of Nb doping, from 3622.90 Oe when x = 0.00 to 4309.17 Oe when x = 0.15. The reasons for this are as follows. First, with the continuous increase of Nb element, the grain size is refined. As is understood, the grain size is the main factor affecting coercivity. Second, the magnetic anisotropy field ($H_a$) is also one of the main factors representing the coercivity of synthetic materials. In this experiment, we found that $H_a$ was reduced, which is obviously inconsistent with this law. However, this is because, after the grain size was refined, the domain walls of large grains could be eliminated, leading to a larger $H_c$. This effect is stronger than the reaction of $H_a$.

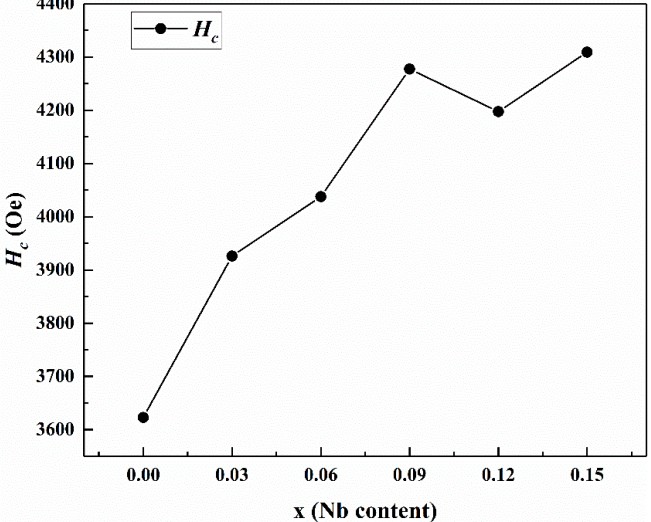

**Figure 10.** Variation of $H_c$ with Nb substitution.

$K_1$ and $H_a$ can be respectively calculated by the following equations [36]:

$$K_1 = M_s \left( \frac{15b}{H^2} \right)^{0.5} \tag{3}$$

$$H_a = \frac{2K_1}{M_S} \tag{4}$$

The experimental magneton number ($n_B$) for the $SrNb_xFe_{12-x}O_{19}$ (x = 0.00–0.15) was deduced using the following expression [24]:

$$n_B = \frac{M.W. \times M_S}{5585} \tag{5}$$

Here, $M_s$ is the saturation magnetization, and *M.W.* is the molecular weight of the sample. The values of $n_B$ are shown in Table 2.

The magneton number $n_B$ here decreased with the increase of Nb doping. The decrease in saturation magnetization is the intuitive reason for the decrease in $n_B$, as the number of magnetons is positively correlated with saturation magnetization [37]. The decrease in $n_B$ was therefore predominantly the result of the weakening of the super-exchange interactions between the different sites in the ferrites.

The SQR ($M_r/M_s$) was calculated from magnetic data, and the values are listed in Table 2. If the squareness ratio is lower than 0.50, there is possibility for the formation of multi-domain structure. Because *SQR* < 0.5, all of the particles were multi-domain structures, and domain walls were present in all of these structures.

## 4. Conclusions

In this article, we successfully synthesized Nb-substituted M-type strontium ferrite and tabulated a number of its properties. Single-phase polycrystalline ferrite can be obtained when the value of x is less than 0.09. FTIR revealed two absorption bands at 450 and 619 $cm^{-1}$, which were attributed to the intrinsic vibrations of tetrahedral and octahedral sites, respectively, in the hexagonal lattice. The Raman spectrum indicated that Nb ions entered the 12k and $4f_2$ orbitals through the peak shift, and all Raman spectrum peaks confirmed the formation of M-type strontium ferrite. SEM images showed that the grain size decreased with the increase of Nb, which played a significant role in grain refinement. Overall, the magnetic properties of the prepared samples were greatly improved. Assuming a small loss of saturation magnetization, the coercive force was greatly improved, and the maximum value of 4309.2 Oe was obtained when x = 0.15. Compared with the undoped increasing coercivity, there is an improvement of 686.3 Oe, or 18.9%. Such an increase is generally difficult to achieve.

**Author Contributions:** W.Z.: Conceptualization, Methodology, Data curation, Writing—original draft preparation, Writing—review & editing. P.L.: Writing—review & editing, Visualization. Y.W.: Investigation. J.G.: Software. J.L.: Resources, Supervision. S.S.: Validation. S.M.: Writing—review & editing; X.S.: Software. All authors have read and agreed to the published version of the manuscript.

**Funding:** This work was supported by the National Natural Science Foundation of China (Grant No. 51764045), the Inner Mongolia Autonomous Region Science and Technology Plan Project (Grant No. 2021GG0438), the Science and Technology Program of Baotou City of China (Grant No. 2019Z3004-5), and the Inner Mon-golia Natural Science Foundation (2020MS05048, 2020BS05029).

**Institutional Review Board Statement:** Not applicable.

**Informed Consent Statement:** Not applicable.

**Data Availability Statement:** Not applicable.

**Acknowledgments:** Special thanks to Yang Yujie for helping me, and my wife, Wang Ying, who is an excellent teacher in Yiyuan No. 1 Middle School in Shandong Province, for her help with the language of the article.

**Conflicts of Interest:** The authors declare that they have no known competing financial interests or personal relationships that could have appeared to influence the work reported in this paper.

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
