# Peer review of "Structure, Spectra, Morphology, and Magnetic Properties of Nb5+ Ion-Substituted Sr Hexaferrites"

_magnetochemistry, doi:10.3390/magnetochemistry8050051_

Round 1

Reviewer 1 Report

Structure, spectra, morphology, and magnetic properties of Nb5+ ion-substituted Sr hexaferrites

Authors: Wenhao Zhang * , Pengwei Li , Yonglun Wang , Jing Guo , Jie Li * , Shuo Shan , Saisai Ma , Xing Suo

Manuscript ID: magnetochemistry-1651844

The present work details the fabrication and the structural and magnetic properties of Nb substituted Sr hexaferrites.  A careful analysis of the fabrication process is indicated through the thermal analysis by TG and DSC measurements. The structure of the samples is measured by X-ray diffraction and Rietveld refinement as well as through FTIR and Raman spectroscopies. The morphology of the samples was revealed by SEM. Magnetic measurements on the samples were performed by VSM.

The manuscript merits for publication in Magnetochemistry although some MAJOR revisions have to be addressed.

  • In the second paragraph of page 3 (3.2 X-ray diffraction analysis) the authors indicate that they calculate “the average crystal grain size (D) as listed in Table 1”. Since this value is particularly large for x=0.06, the authors should indicate the road to perform this calculation.
  • In Table 1, the authors introduce the parameter c2 without any previous mention in the main text of the manuscript. The authors should indicate the physical magnitude that the c2 parameter is referring to.
  • In Table 2, the authors indicate the results of the elemental analysis of the composition of the Nb substituted ferrites. Previously, the authors stated in section 3.5 (Morphological analysis) that “no other elements were detected in any of the samples, confirming the purity of the material”. However, in Table 2, C appears in every sample. The authors should indicate the reason of the presence of C in all the samples.
  • In addition, in Table 2, Nb is also present in the SrFe12O19 ferrite sample. The authors should explain this contradictory result.
  • The authors should clearly indicate whether the values appearing in Table 2 refer to mass concentration, atomic concentration or any other magnitude not mentioned in the main text of the manuscript. They should also check whether the results of this analysis coincides with the stechiometric formula SrNbxFe12-xO19 of the different samples.
  • Equations (2) and (3) do not appear in reference [25] as indicated by the authors. They should indicate the correct reference where equations (2) and (3) were obtained from.
  • The authors indicate in the fourth paragraph of section 3.6 (Magnetic property analysis) that “the magnetocrystalline anisotropy constant (Ha)”. Typically Ha refers to the magnetic anisotropy field, which is related to the magnetocrystalline anisotropy energy density K. The authors correlate both magnitudes in equation (3). The authors should use the correct definitions in the amended version of the manuscript.
  • From the Stoner-Wohlfarth model, equation (3) is valid for a set of aligned particles with uniaxial anisotropy. This does not seem to be the situation, according to the hysteresis loop shown in Figure 8. These hysteresis loops would rather correspond to a set of particles whose anisotropy axis were randomly distributed. Thus, equation (3) would not be valid and the authors should indicate the correct expression correlating anisotropy energy density and anisotropy field.

Minor revisions.

  • Finally, it would be convenient for the authors to include an inset in Figure 8, amplifying the horizontal scale to show more clearly the value of the coercivity of the different samples.

Reviewer 2 Report

This work presents the structure, spectra, morphology, and magnetic properties of Nb5+ ion-substituted Sr hexaferrites. The authors present results which demonstrated that the coercivity increased with increases in the doping amount.This effect is investigated deeply and the conclusions are supported by the experimental data. However, some suggestions should be considered by the authors.

1) The authors must give more information about the SEM images. They should try to adjust the sizes of the SEM images. To make the comparison possible, images have to be chosen with the same magnification but in the same time the scale has to be the original one. At the present form, all conclusions derived from these images are speculative.

2) The authors should make statistics for particles size distribution from more SEM images for samples with different doping amount.

3) The authors claim that "after the grain size was refined, the domain walls of large grains could be eliminated". What is the critical size of the grains for monodoman state for this material?

Round 2

Reviewer 1 Report

Structure, spectra, morphology, and magnetic properties of Nb5+ ion-substituted Sr hexaferrites

Authors: Wenhao Zhang * , Pengwei Li , Yonglun Wang , Jing Guo , Jie Li * , Shuo Shan , Saisai Ma , Xing Suo

Manuscript ID: magnetochemistry-1651844

The authors have modified the manuscript according to the reviewer’s suggestions but still some experimental results have to be clearly explained in order to improve the quality of the manuscript.

Again, the manuscript merits for publication in Magnetochemistry although some MAJOR revisions have to be addressed.

  1. The main contradiction in this manuscript originates from Table II. Still, the authors do not indicate in Table II whether the results correspond to atomic or weight element content concentration. According to the EDS spectrum that the authors provided in the reply to the reviewer’s comments, it seems to me that the results in Table II correspond to weight concentration (Wt%, in the EDS spectrum). From this weight concentration the authors could calculate the atomic concentration and check whether this concentration from EDS spectra coincides with the assumed stoichiometric formula of the ferrites in the present manuscript.
  2. Another contradiction seems to arise from the amount of Nb in the x=0.00 sample. In the first version of the manuscript the authors found a finite amount of Nb in the x=0.00 sample. This amount of Nb seems to arise from the peak in the 2.2 channel in the EDS spectrum of the x=0.00 sample (provided by the authors in the reply letter). In this spectrum, this peak is not labelled (and thus, it is not considered in the final formula in the inset of the upper right corner of the graph), but in the rest of the spectra of the other samples this 2.2 peak is labelled as Nb and consequently taken into account in the element weight percentage of the formula. The authors should clarify whether the peak in channel 2.2 corresponds to Nb, to any other element, or whether this is a systematic error.
  3. As a final remark, the results of the weight or atomic concentration from the EDS spectrum are sometimes semiquantitative estimations. The authors could just indicate that the EDS test data indicate the increase in the amount of Nb throughout the different samples of the series and remove Table II that could lead to contradictory results. The rest of the data presented in the manuscript are worth publishing in Magnetochemistry.

Round 3

Reviewer 1 Report

The authors have modified the manuscript according to the reviewer's suggestions. The manuscript might be publish in Magnetochemistry.